# Multisensory integration enhances audiovisual responses in the Mauthner cell

Santiago Otero-Coronel[1], Thomas Preuss[2], Violeta Medan[1,3]*

[1]Instituto de Fisiología y Biología Molecular y Celular, Consejo Nacional de Investigaciones Científicas y Tecnológicas, Buenos Aires, Argentina; [2]Department Psychology, Hunter College, City University of New York, New York, United States; [3]Department Fisiología y Biología Molecular y Celular, Facultad de Ciencias Exactas y Naturales, Universidad de Buenos Aires, Buenos Aires, Argentina

## eLife Assessment

This study provides **valuable** advances in our understanding of how inputs from multiple sources can impact the physiology of motor neurons during the process of multisensory integration. Specifically, the authors show how streams of auditory and principally visual information modulate the physiology of Mauthner neurons in goldfish, thus allowing the different senses to influence escape behavior. Supporting evidence is generally **convincing**, although material reporting the direct control of behavior is less representative of the data.

*For correspondence:
violetamedan@fbmc.fcen.uba.ar

Competing interest: The authors declare that no competing interests exist.

**Abstract** Multisensory integration (MSI) combines information from multiple sensory modalities to create a coherent perception of the world. In contexts where sensory information is limited or equivocal, it also allows animals to integrate individually ambiguous stimuli into a clearer or more accurate percept and, thus, react with a more adaptive behavioral response. Although responses to multisensory stimuli have been described at the neuronal and behavioral levels, a causal or direct link between these two is still missing. In this study, we studied the integration of audiovisual inputs in the Mauthner cell, a command neuron necessary and sufficient to trigger a stereotypical escape response in fish. We performed intracellular recordings in adult goldfish while presenting a diverse range of stimuli to determine which stimulus properties affect their integration. Our results show that stimulus modality, intensity, temporal structure, and interstimulus delay affect input summation. Mechanistically, we found that the distinct decay dynamics of FFI triggered by auditory and visual stimuli can account for certain aspects of input integration. Altogether, this is a rare example of the characterization of MSI in a cell with clear behavioral relevance, providing both phenomenological and mechanistic insights into how MSI depends on stimulus properties.

## Introduction

The ability to detect a potential threat and react with an escape behavior is critical for survival. In nature, however, many predators develop strategies such as camouflage or partial hiding to decrease their salience, and sensory stimuli tend to be noisy and ambiguous. Consequently, it is adaptive for animals to have ways to disambiguate the signals from the sensory inflow they receive, especially in contexts where available information is limited.

One strategy for reducing sensory uncertainty is multisensory integration (MSI), which is defined as the process of combining information coming from multiple sensory streams to form a new percept

with reduced ambiguity (*Stein and Stanford, 2008*). MSI has been extensively documented in animals from nematodes to humans and in a broad range of behavioral contexts (*Wallace et al., 1998*; *Metaxakis et al., 2018*; *Zhou et al., 2019*; *Gil-Guevara et al., 2022*). In the context of escape behaviors, MSI of threat-like stimuli increases the escape probability and reduces the latency of the evoked response (*Rowland et al., 2007*; *Martorell and Medan, 2022*). In mammals, activation of different parts of the superior colliculus determines whether animals engage in exploratory or defensive behaviors (*Dean et al., 1989*; *Wei et al., 2015*). Importantly, the superior colliculus contains neurons that integrate multisensory stimuli due to synaptic convergence of afferents conveying information from different sensory modalities (*Kadunce et al., 2001*). These studies established that the effects of MSI are larger when weak or ambiguous stimuli are combined, thus showing an inverse effectiveness relationship with stimulus strength. The underlying hypothesis is that integration of sensory signals from different modalities form a more accurate world percept only if each unisensory signal carries some uncertainty or ambiguity.

In goldfish, audiovisual MSI increases the probability and reduces the latency of startle escape behaviors, with an inverse relationship between the MSI effectiveness and the stimulus salience (*McIntyre and Preuss, 2019*; *Martorell and Medan, 2022*). These startle escapes (C-starts) are triggered by a pair of reticulospinal neurons (Mauthner cells) that receive visual and auditory inputs through anatomically segregated pathways. Auditory inputs reach the M-cell lateral dendrite through a disynaptic (2 ms) pathway, while the visual inputs relay information in the optic tectum before reaching the M-cell ventral dendrite through a polysynaptic (20 ms) pathway (*Zottoli et al., 1987*; *Pereda et al., 2004*; *Korn and Faber, 2005*; *Szabo et al., 2007*). Interestingly, the auditory and visual M-cell dendrites show differences in their cable properties, which appear to be well suited to process inputs from abrupt (i.e. fast raise time) auditory pips and more gradually raising visual looms (*Medan et al., 2018*). However, it is unclear how auditory and visual signals are integrated at the level of the M-cell to generate a multisensory response.

The accessibility of M-cells for in vivo intracellular recordings provides an excellent opportunity to investigate the cellular mechanisms of MSI. Moreover, the one-to-one relationship between a single M-cell action potential and the initiation of a C-start (*Zottoli, 1977*; *Weiss et al., 2006*; *Zwaka et al., 2022*) provides a clear link between behavior and the observed integration in the M-cell. As such, the goal of this study was to reveal the cellular mechanism that underlie the multimodal integration of M-cell initiated startle escapes we observed in the behavioral experiments (*McIntyre and Preuss, 2019*; *Martorell and Medan, 2022*). Specifically, we studied how visual and auditory presynaptic pathways and intrinsic membrane properties of the M-cell interact to produce MSI to ultimately enhance threat detection. To this end, we performed intracellular in vivo recordings from the M-cell soma using auditory and tectal stimuli with naturalistic dynamics (*Preuss et al., 2006*; *Szabo et al., 2006*; *Medan et al., 2018*). Our results show that the temporal dynamics and strength of stimuli, as well as sensory evoked feedforward inhibition (FFI) affect the magnitude of MSI. The results represent a rare characterization of MSI at a single-cell level in a neuron that is sufficient and necessary to trigger a distinct behavior.

## Results

### Mauthner cell responses are sensitive to the dynamics of tectal stimuli

MSI is known to be dependent on the saliency of the unimodal components (*Stanford et al., 2005*; *Martorell and Medan, 2022*). To create a stimulus set of variable strength and temporal dynamics, we recorded M-cell responses to different electrical trains delivered in the posterior rim of the tectum. As shown in other species, goldfish adapt rapidly to repeated natural-looming stimulation (*Marquez-Legorreta et al., 2022*; *Fotowat and Engert, 2023*). Indeed, in previous behavioral experiments we presented looming stimuli with intertrial intervals (ITI) of about 4–5 min to reduce such adaptation (*Preuss et al., 2006*; *Otero Coronel et al., 2020*; *Martorell and Medan, 2022*). However, ITI of 4–5 min are prohibitively long for steady intracellular recordings. Considering these restrictions, we used tectal stimuli (that bypass the retina) at an ITI of 30 s which does not produce adaptation. This tectal stimulation results in depolarization responses in the ventral (visual) dendrite of the M-cell that are qualitatively and quantitatively similar to those elicited by visual looming stimuli. While this approach bypasses retinal and tectal processing of visual inputs, it is important to note that the

tectum also integrates multisensory inputs, including from the thalamus and other sensory pathways such as the auditory and vestibular systems (*Bastian, 1982*; *Finger and Tong, 1984*; *Fame et al., 2006*; *Kardamakis et al., 2016*; *Thompson et al., 2016*; *Poulsen et al., 2021*). As a result, tectal stimulation is likely to recruit projections that are predominantly, but not exclusively, visual. Nonetheless, tectal afferents reach the distal ventral dendrite of the M-cell and produce non-adapting synaptic responses that retain key characteristics of looming responses such as fast excitatory post-synaptic potentials (EPSPs) riding on top of a ramped depolarization (*Preuss et al., 2006*; *Szabo et al., 2006*; *Medan and Preuss, 2014*; *Medan et al., 2018*; *Zottoli et al., 1987*, *Figure 1A*). M-cell somatic responses reach a plateau after 30–100 ms and decay to resting values 200–300 ms after the end of the tectal train. To characterize how the M-cell responses depend on the frequency and duration of the stimulation trains in the tectum, we recorded responses to 60 Hz trains that varied in duration (*Figure 1A, B*, N = 18), and 100 ms tectal trains that varied in pulse frequency (*Figure 1C, D*, N = 11). The responses to both sets of trains exhibited complex temporal dynamics with a tonic component that progressively builds during the train and a fast phasic component after each pulse (*Figure 1A'*).

Increasing the duration of the 60 Hz trains drove stronger tectal responses due to the increase of the tonic component of the response while the phasic response remained unchanged (ANOVA, tonic: $F(3,40) = 37.29$, p < 0.0001, phasic: $F(3,40) = 0.58$, p = 0.6311, *Figure 1B*). Increasing the frequency of the 100 ms stimulation trains also drove stronger tectal responses due to a large increase in the tonic component (ANOVA, tonic: $F(3,30) = 12.29$, p < 0.0001). However, increasing the tectal stimulus frequency produced a significant reduction in the amplitude of the phasic component (ANOVA, phasic: $F(3,30) = 11.29$, p < 0.0001, *Figure 1D*). Comparing the stimulus trains of variable duration and frequency reveals that modifying the duration of the 60 Hz trains or the frequency of 100 ms trains is similarly effective in tuning the magnitude of the responses. Increasing the duration of a 60 Hz from 33 to 200 ms produced a 45% increase in the total response amplitude (phasic + tonic) while increasing the frequency from 30 to 200 Hz produced a 50% increase in the response amplitude (*Figure 1B, D*). Since multimodal integration is known to depend on the strength of the unimodal components (*Stanford et al., 2005*; *Martorell and Medan, 2022*), in the following multimodal experiments we leveraged this characterization to generate tectal stimuli of variable strength and dynamics. Although both the 60 Hz trains of variable duration and the 100 ms trains of variable frequency evoked responses of similar total amplitude, we decided to include both sets of trains in our multisensory stimuli to determine whether any detected effect depended on the temporal structure of the stimulus or the ratio of phasic-to-tonic responses.

One of the principles that guide MSI is the spatial coherence of the different cues being integrated. If the perceived source localizations of two stimuli overlap, these stimuli are more likely to be integrated. Conversely, if the two stimuli arrive from spatially disparate sources the chances of observing an MSI will decrease (*Meredith and Stein, 1996*; *Stein and Stanford, 2008*). The geometrical configuration of the setup in our experiments determined that the loudspeaker used to produce acoustic stimuli was mounted to the right of the fish and electrical stimulation was applied to the left tectum (which normally receives visual signals from the right retina). Whenever possible, we performed recordings from both Mauthner cells. To rule out the possibility of a systematic difference in the responses of either M-cell due to the spatial source of the stimuli, we compared the responses to tectal or auditory stimuli in sequential recordings of the right and left M-cells of the same fish. *Figure 1—figure supplement 1* shows representative trace examples for tectal (100 ms 60 Hz tectal train, *Figure 1—figure supplement 1A*) and auditory (5 ms 200 Hz auditory pip, *Figure 1—figure supplement 1B*) stimulation from the left (upper traces) or right (lower traces) M-cell of the same fish. The complex post-synaptic potentials were highly stereotypical in successive presentations of tectal or auditory stimuli within and across cells. Analysis from pairs of tectal (*Figure 1—figure supplement 1C*, N = 9) and auditory (*Figure 1—figure supplement 1D*, N = 11) M-cell responses showed no statistical differences between ipsi- and contralateral stimulation (paired *T*-test, T: $t(8) = 0.18$, p = 0.8573, A: $t(10) = -0.76$, p = 0.4672). This confirms previous results indicating that in goldfish, tectal stimulation produces responses with similar latency and amplitude in both M-cells (*Zottoli et al., 1987*; *Canfield, 2006*). We therefore pooled data acquired from left or right M-cells for all following analyses.

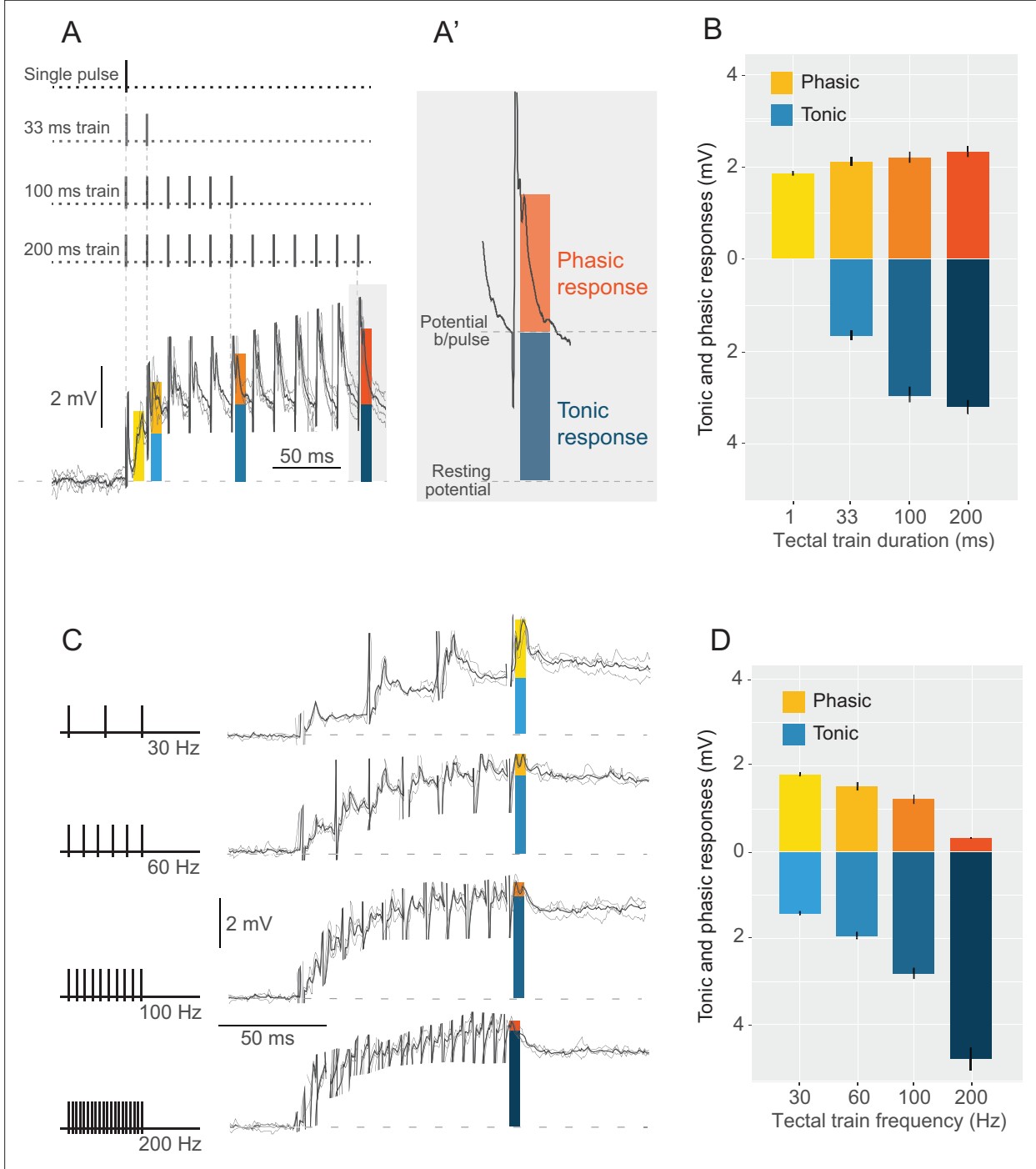

**Figure 1.** M-cell responses to tectal stimulation. (A) M-cell responses to a 60-Hz tectal train of 200 ms were used to measure the evoked depolarization at 1, 33, 100, and 200 ms after stimulus onset during a 12-ms time window (colored vertical boxes). (A') Blue and orange bars in the trace show the tonic and phasic components of the response, respectively. The horizontal dotted lines indicate the baseline (bottom) and the potential before the last pulse for quantification of each component. (B) Mean amplitude (± SEM, N=18) of the phasic (shades of orange) and tonic (shades of blue) components of tectal trains for the time windows described in A. (C) M-cell responses to a 100-ms tectal pulse train of 30, 60, 100, or 200 Hz were recorded and quantified as in A, at the end of the pulse. (D) Mean amplitude (± SEM, N=11) of the phasic (shades of orange) and tonic (shades of blue) components of 100 ms tectal trains of different frequencies.

The online version of this article includes the following figure supplement(s) for figure 1:

**Figure supplement 1.** Responses in the left- and right-side Mauthner cells.

## Multisensory enhancement in the Mauthner cell

Abrupt sound stimuli have been extensively used to trigger the short latency escape responses initiated by firing of the M-cell (*Eaton et al., 1977*; *Burgess and Granato, 2007*; *Neumeister et al., 2008*; *Weiss et al., 2009*; *Zheng and Schmid, 2023*). Acoustic stimulation produces a complex response in the M-cell which is composed of a fast EPSP superimposed on a depolarizing envelope termed slow EPSP (*Szabo et al., 2006*). The fast EPSP is composed of fast electrical coupling potentials resulting from firing of the eighth nerve auditory afferences that synapse the distal end of the lateral dendrite of the M-cell (club endings) (*Pereda et al., 2004*; *Szabo et al., 2007*). In contrast, the slow EPSP is originated by chemical (glutamatergic) synaptic contacts on the soma and proximal dendrite conveying information from inner hear endorgans and it has been shown that it mostly carries information on stimulus intensity (*Szabo et al., 2006*). Although the M-cell is known to respond to stimuli of different modalities, there are no reports on how it combines and integrates information when auditory and tectal sources are co-presented. To determine whether multisensory enhancement occurs in the M-cell, we recorded somatic responses to an auditory pip (unisensory auditory, A), a tectal stimulation train (unisensory tectal, T), and their combination (multisensory audiotectal, M) (*Figure 2A, E*).

Increasing the duration of the 60 Hz tectal train positively modulated the multisensory responses (amplitudes ranging from 3.08 ± 0.84 to 5.29 ± 1.64 mV, $N$ = 17, $n$ = 47; LM: evoked depolarization ~ tectal duration, $F(1,45)$ = 17,39, p = 0.0001, *Figure 2B*). We then compared the multisensory responses to their unisensory counterparts. When the multimodal responses were normalized to the most effective unisensory response, the resulting MSI/Max indices (see Methods), ranged from 1.48 to 1.53 across all stimulus durations (Wilcoxon single sample test mu = 1, alternative mu ≠ 1, p < 0.001 for all stimulus durations, *Figure 2C*), indicating a significant multisensory enhancement of the response. When the multimodal response was normalized to the sum of both unisensory responses, the resulting MSI/Sum indices (see Methods) were less than 1 (i.e. sublinear integration, Wilcoxon single sample test mu = 1, alternative mu ≠ 1, p < 0.01) for all cases except for the briefest tectal stimulus, for which the MSI/Sum index was 1.06 (i.e. linear integration, Wilcoxon single sample test mu = 1, alternative mu ≠ 1, p < 0.0676, *Figure 2D*).

Increasing the frequency of a 100-ms tectal train (*Figure 2E*) was not as effective in modulating the multisensory responses (LM: evoked depolarization ~ tectal frequency, $F(1,39)$ = 3.96, p = 0.0535, *Figure 2F*). However, the multimodal responses still exhibited MSI/Max indices that ranged 1.43–1.53 when compared to the largest unimodal stimulus (*Figure 2G*) indicating that a 100-ms tectal train (irrespective of its frequency) significantly enhances the M-cell response. The MSI/Sum of these multimodal responses ranged 0.82–0.86 showing a similar sublinear integration to that of the multisensory stimuli with the longer 60 Hz trains (Wilcoxon single sample test mu = 1, alternative mu ≠ 1, all p ≤ 0.01, *Figure 2H*). Neither MSI/Max nor MSI/Sum showed a significant dependency with tectal frequency (LM: MSI/Max ~ tectal frequency, $F(1,39)$ = 0.02, p = 0.882, MSI/Sum ~ tectal frequency, $F(1,39)$ = 0.49, p = 0.487).

Overall, these results show that, in all cases, combining an auditory pip with a preceding tectal pulse enhances the M-cell response by about 50%. This multisensory enhancement does not show a strong modulation by the duration or frequency of the tectal stimulus (*Figure 2C, G*). However, while most multisensory stimuli exhibited a sublinear integration, the multisensory stimulus with the shortest (and weakest) tectal component produced a linear response (*Figure 2B, D*).

## Tectal- and auditory-evoked FFIs exhibit distinct temporal dynamics

Both visual and auditory stimuli evoke FFI at the perisomatic region of the M-cell through a population of inhibitory interneurons which open chloride channels that produce shunting of the postsynaptic currents (*Diamond et al., 1973*; *Zottoli et al., 1987*; *Koyama et al., 2011*; *Tabor et al., 2018*). This FFI restricts the temporal window of sensory excitation to terminate the excitatory signal. In combination with passive membrane decay, an increase in FFI will further narrow the window for temporal integration. Conversely, a decrease in FFI would increase the lingering depolarization of a response produced by a previous stimulus, and thus extend the temporal integration window. These differences in the strength or temporal dynamics between visual- and auditory-evoked FFI may affect MSI (*Felch et al., 2016*). We used a single auditory pip and a brief tectal train as 'conditioning' stimuli of similar strength, which evoked comparable peak depolarizations in the M-cell (~5 mV) (*Figure 3A*). Peak FFI inhibition induced by auditory and tectal stimulation showed similar amplitude (*Figure 3B*, auditory

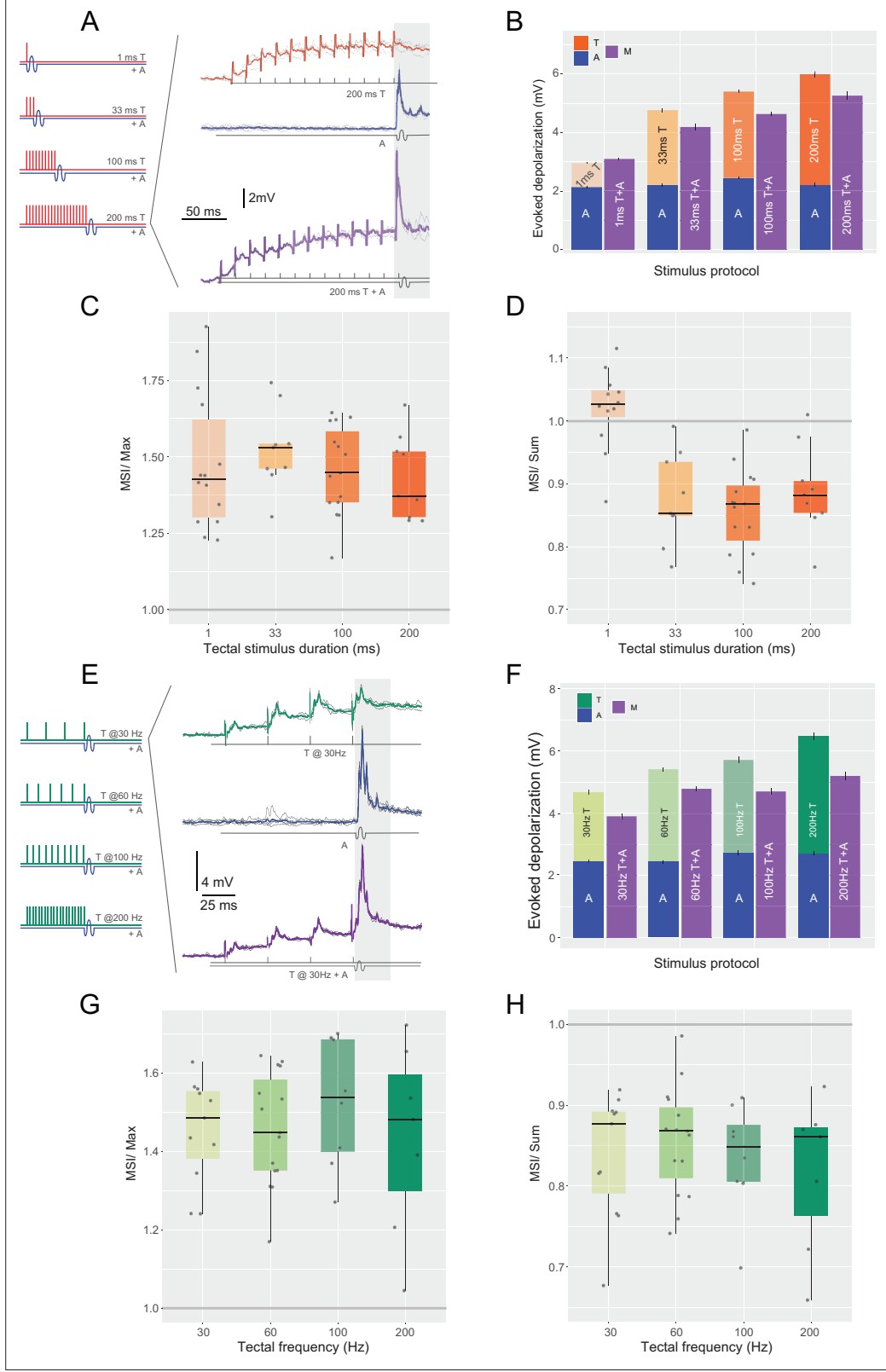

**Figure 2.** Multisensory integration in the M-cell. (**A**) Diagram of stimulation (left) and response examples (right). The response examples correspond, to a single acoustic pulse (blue trace) and to their multisensory combination (purple). Below the traces diagram of the stimulation is shown in gray. (**B**) Stacked bars comparing mean (± SEM, N=17) depolarization evoked by tectal trains of indicated duration (T, orange), auditory (A, blue), or their

multisensory combination (purple). (**C**) Boxplots showing the ratios between the multisensory responses and the maximum unisensory response (MSI/Max). (**D**) Boxplots showing the ratios between the multisensory responses and the sum of its unisensory components (MSI/Sum). (**E–H**) As in A–D but for 100 ms tectal trains of different frequencies.

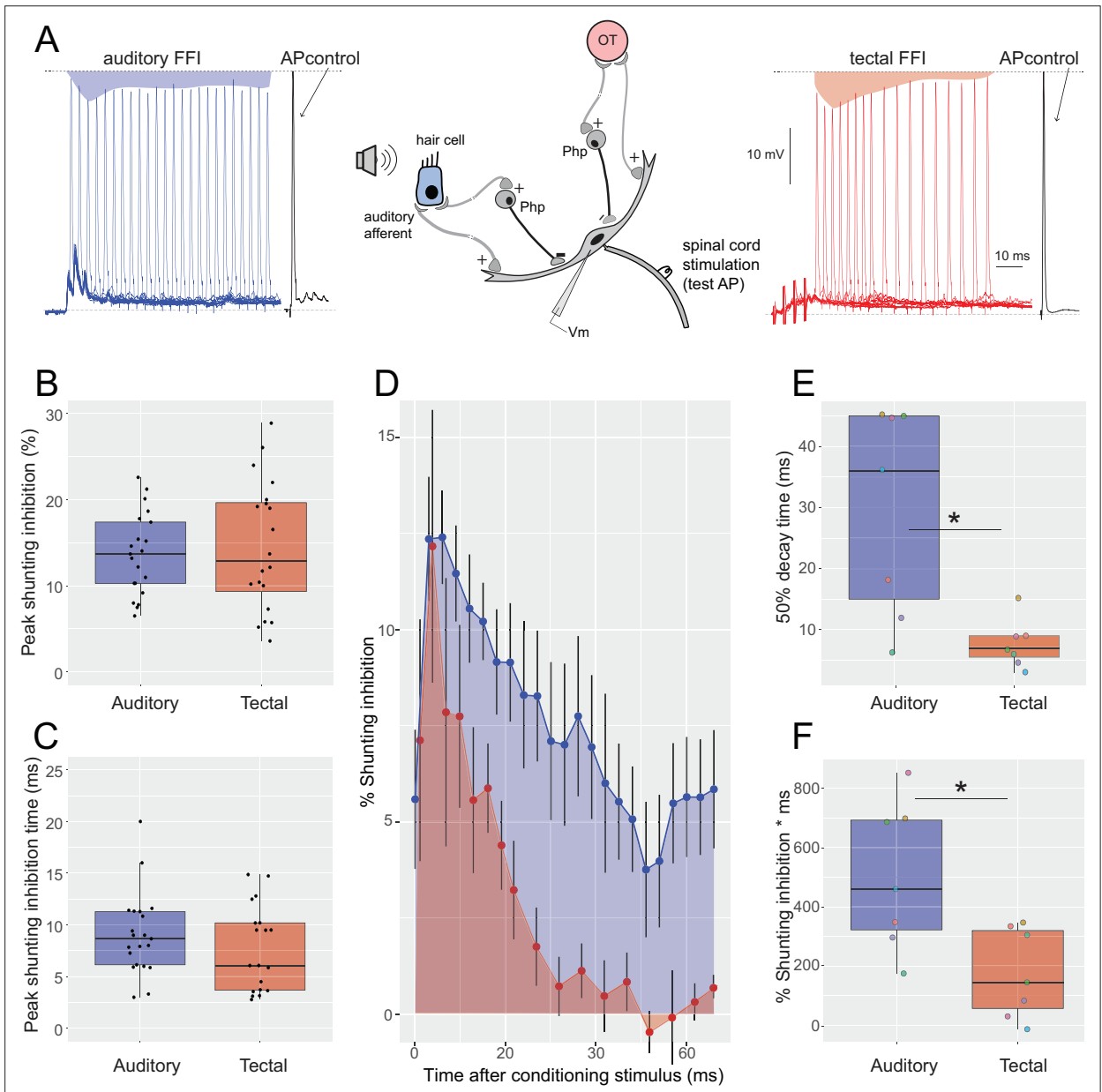

**Figure 3.** Auditory and tectal feedforward inhibition (FFI) have different decay dynamics. (**A**) Diagram of auditory or tectal FFI circuits (center), and the quantification of sensory evoked shunting inhibition as a reduction of M-cell AP amplitude (left, auditory: shaded-blue area; right, tectal: shaded-red area) to a control AP (black trace). (**B, C**). Boxplots of peak inhibition and time of peak inhibition triggered by acoustic or tectal stimuli. (**D**) Mean (± SEM, N=19) FFI triggered by tectal and auditory inputs measured in the same M-cells at different time points between 0 and 70 ms post-stimulus. (**E**) Boxplot of the time elapsed to 50% decay of peak FFI for the auditory (left) and tectal (right) stimuli. (Kruskal–Wallis chi-squared, p = 0.0145). (**F**) Area below the curve (0–70 ms) for the auditory (left) and tectal (right) stimuli. (Kruskal–Wallis chi-squared, p = 0.0181).

vs. tectal %SI: 13.65 ± 4.74 vs. 14.55 ± 7.50; Kruskal–Wallis chi-squared (1) = 0.01, p = 0.9066, *N* = 19) with peak inhibition at the same time (auditory vs. tectal time (ms) of maximum %SI: 9.04 ± 3.93 vs. 8.85 ± 6.54; Kruskal–Wallis chi-squared (1) = 0.30, p = 0.5838, *Figure 3C*, N = 19). In a subset of experiments (*N* = 7), both auditory and tectal FFI were evaluated sequentially in the same cell and same recording site which allowed us to have an accurate estimate of the inhibition triggered by each sensory modality (*Figure 3D*). Although the maximum level of inhibition was comparable and was reached in a similar time, recovery from inhibition was much faster for tectal than for auditory FFI. Tectal FFI decayed to 50% of its peak in 8 ms, compared to 30 ms for the auditory FFI (Kruskal–Wallis chi-squared (1) = 5.97, p = 0.0145, *Figure 3E*). In addition, the area under the curve for auditory FFI was almost three times greater than for tectal FFI (502 vs. 176%*ms, respectively, Kruskal–Wallis chi-squared (1) = 5.58, p = 0.0181, *Figure 3F*).

These results highlight the key role of the temporal dynamics of the inhibition that reaches the M-cell and which in turn will affect integration in the M-cell. Moreover, these results show that auditory stimuli trigger more prolonged inhibition than tectal stimuli of similar amplitude and decay time. The short-lived FFI that follows tectal stimulation suggests that multiple trains or waves of tectal inputs are more likely to be integrated and summed than multiple sounds. This goes in line with the duration of visual looms that are most effective in triggering the M-cell and eliciting visual C-start escapes (*Preuss et al., 2006*; *Temizer et al., 2015*). Differences on the decay rate of inhibition produced by brief tectal or auditory stimuli could affect integration of pairs of brief sensory stimuli depending on their sequence and their timing, which is explored in the next section.

## Stimulus temporal order and modality affect stimuli integration

There are several anatomical and functional differences in the pathways followed by auditory and visual inputs that eventually reach the M-cell soma. First, auditory inputs from the eighth nerve reach the distal lateral dendrite of the M-cell through a disynaptic (≈2 ms) pathway followed by slower inputs from glutamatergic synaptic contacts on the soma and proximal dendrite. Visual inputs reach the ventral dendrite after a polysynaptic (≈20 ms) pathway that includes the optic tectum (*Zottoli*

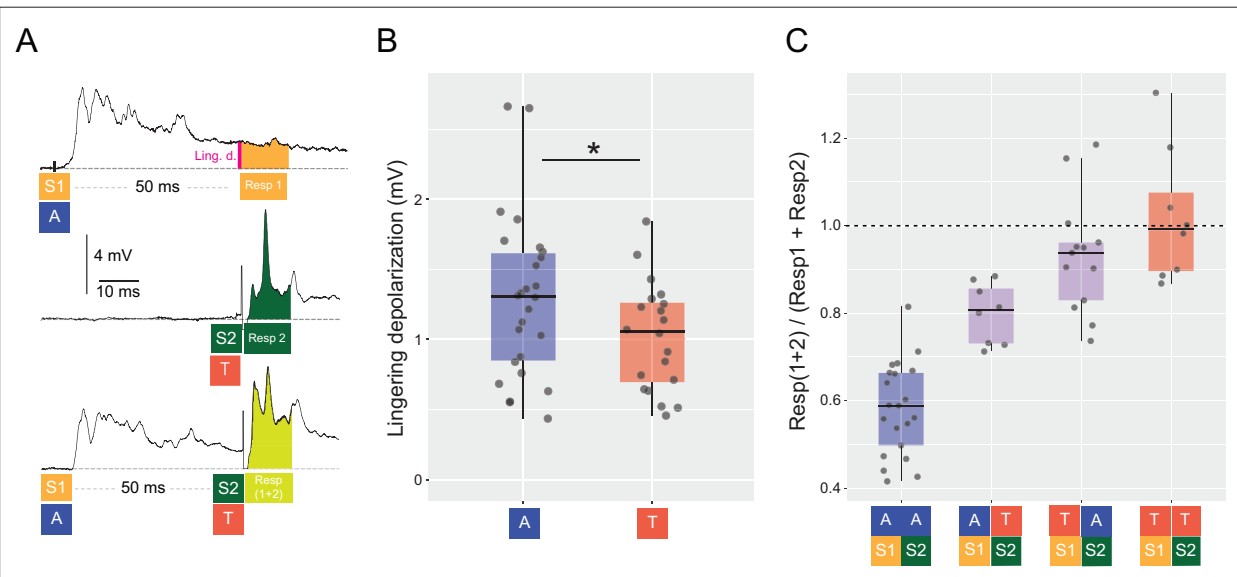

**Figure 4.** Effect of stimulus sequence and modality on M-cell integration. (**A**) Representative responses of one M-cell to two stimuli presented in different trials, or in the same trials with a 50-ms offset. The upper trace shows the M-cell response to an auditory stimulus (S1/A), middle trace corresponds to the response to a single tectal pulse (S2/T), and bottom trace shows the response to an auditory stimulus followed by a tectal stimulus 50 ms later. The area used to quantify each response is shaded in orange (Resp1), dark green (Resp2), or light green (Resp(1 + 2)), respectively. Schematics below each trace the order of stimuli and the area where responses were quantified. The pink bar indicates the time point at which the lingering depolarization of the first response was measured. Note that this analysis was performed for trials in which S1 and S2 could be either auditory or T, resulting in trials AA, TT, TA, and AT sequences. (**B**) Lingering depolarization after 50 ms of a single auditory pip or a short tectal train. (Kruskal–Wallis rank sum test, p = 0.0188). (**C**) Integration in the M-cell for pairs of uni- or multisensory stimuli with a 50-ms delay was calculated as the ratio: Resp(1 + 2)/(Resp1 + Resp2).

*et al., 1987*; *Szabo et al., 2006*). Second, M-cell dendritic characteristics determine modality specific filtering differences resulting in stronger attenuation of visual than auditory signals (*Medan et al., 2018*). For auditory stimuli, it has been reported that a sound pip that precedes by 20–150 ms a second sound reduces the M-cell response to the second sound, a phenomenon known as auditory prepulse inhibition (*Neumeister et al., 2008*; *Curtin and Preuss, 2015*; *Tabor et al., 2018*). However, it is unclear whether a similar inhibitory integration would hold true for cross-modal and tectal–tectal interactions, since auditory and visual inputs travel through distinct and separated pathways and undergo different filtering.

To investigate the role of input modality on M-cell integration we used pairs of similarly brief (~10 ms) auditory pips and tectal pulses 50 ms apart with in all possible sequences, including unisensory combinations (AA, TT) and multisensory combinations (AT, TA, *Figure 4*, N = 29, n = 50). We decided to use a 50-ms interval because it was observed to be a period after which the tectal FFI had returned to baseline values, while the auditory FFI remained at about 4%. The amplitude of the tectal stimulation pulse was chosen so that the M-cell responses to the tectal and auditory stimuli were of similar magnitude (4–8 mV). The initial response to the first stimulus (S1, auditory in the example shown in *Figure 4A*, top trace) decayed passively over the course of 50 ms, and at that point the second stimulus was presented (S2, tectal in the example shown if *Figure 4A*, bottom trace). The mean lingering depolarization remaining immediately before the second stimulus was delivered (*Figure 4A*, top trace, pink bar) was slightly larger for the evoked auditory pip than for the tectal stimulus (*Figure 4B*, A: 1.48 ± 0.79 mV vs. T: 0.95 ± 0.50 mV, Kruskal–Wallis rank sum test (1) = 5.5173, p = 0.0188).

We next calculated the integration indices of the four possible combinations of uni and multisensory pairs (AA, AT, TA, and TT). For that, we measured the depolarization in A-only or T-only trials in 12 ms windows. These windows started either 50 ms after stimulus onset to account for the contribution of the first stimulus, S1 (*Figure 4A*, Resp1, orange area of top trace), or immediately after stimulus presentation to account for the contribution of the second stimulus, S2 (*Figure 4A*, Resp2, green area of middle trace). Then, we measured the response for pairs of stimuli in a 12-ms window after the presentation of S2 (*Figure 4A*, Resp (1 + 2), bottom trace). Finally, we calculated the integration for each type of uni- (AA and TT) or multisensory stimuli (AT and TA) defining a 'S1–S2 Integration Index' as S1–S2 Int./Sum = Resp(1 + 2)/(Resp1 + Resp2). Note that for multisensory trials (i.e. AT or TA), this metric is equivalent to the MSI/Sum index, except for a 50-ms temporal offset in the presentation and quantification of the first stimulus.

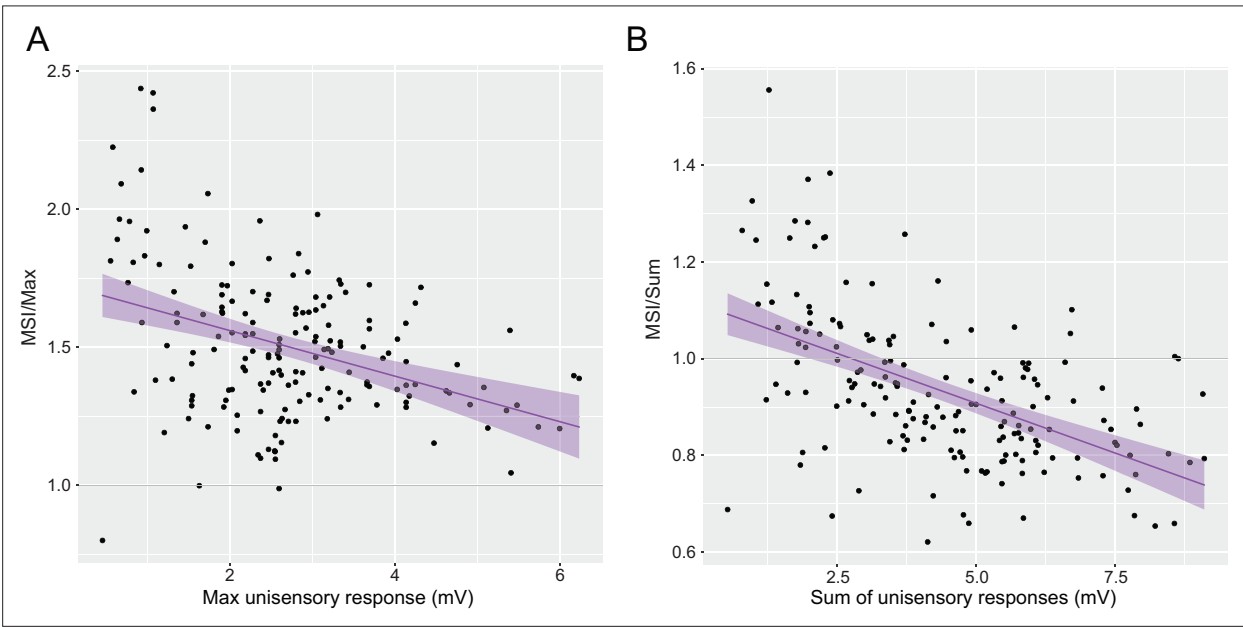

**Figure 5.** Multisensory integration of subthreshold unisensory stimuli. (**A**) MSI/Max vs. the maximum unisensory response of different multisensory stimuli, irrespective of duration, frequency, or sequence of stimulation (Linear Model, p < 0.0001). (**B**) Similar to A, but with the MSI/Sum instead vs. the sum of the unisensory responses (Linear Model, p < 0.0001).

In the AA trials (*N* = 21), we observed an inhibitory integration (S1–S2 Int./Sum < 1, two-sided Wilcoxon rank test, p < 0.0001) with the first auditory pip reducing the amplitude expected for the second pip by 42%, as others have previously described (*Neumeister et al., 2008*; *Curtin and Preuss, 2015*; *Tabor et al., 2018*). The AT combination also showed sublinear integration (S1–S2 Int./Sum < 1, two-sided Wilcoxon rank test, p = 0.0078) while the TT and TA combinations showed linear integration (S1–S2 Int./Sum = 1, two-sided Wilcoxon rank test, VA: p = 0.0681, VV: p = 0.9453, *Figure 4C*). The inhibitory effect in the AA combination was significantly larger than for the AT combination (*Figure 4*, 1w-ANOVA, *F* = 38.34, p < 0.0001).

## MSI in the Mauthner cell shows inverse effectiveness

The principle of inverse effectiveness states that the effects of MSI are typically larger when the components of a multisensory stimulus are weak. Throughout our experiments, we only observed linear integration of multisensory stimuli when presenting single tectal pulses (*Figure 2C*, 1 ms tectal), or brief (~10 ms) trains (*Figure 4*, AT–TA), and observed sublinear integration for multimodal stimuli with longer tectal trains (*Figure 2D–H*). However, these results represent the average MSI for each group for a specific set of experimental conditions and effectively simplify the diverse range of responses across trials, experimental conditions, and fish. To compare the overall dependence of the multisensory enhancement on the strength of sensory inputs, we pooled the MSI indexes for experiments irrespective of sequence of stimuli (AT and TA, *N* = 18, *n* = 174), tectal train duration (from a single 1 ms tectal pulse to a 200 ms train) and frequency (from a single pulse to 1000 Hz) (*Figure 5A*, MSI/Max and B, MSI/Sum). We found a significant inverse correlation between the magnitude of the multisensory enhancement and the magnitude of the unisensory components (MSI/Max, LM, $F(1,172)$: 28.17, p < 0.0001; MSI/Sum, LM, $F(1,172)$: 67.33, p < 0.0001). This result parallels the inverse effectiveness reported at the behavioral level in goldfish (*McIntyre and Preuss, 2019*; *Martorell and Medan, 2022*), confirming that the relative multisensory enhancement obtained by combining two inputs is proportionally higher when the stimuli are weak. Combinations of weak stimuli (<2.5 mV, *Figure 5B*) produce supralinear multisensory responses that are between 50–100% stronger than the most effective unisensory stimulus (*Figure 5A*). Noteworthy, these results provide robust evidence that in the escape network of fish, inverse effectiveness of MSI is implemented, at least, at the single (Mauthner) cell level.

## Discussion

The Mauthner cell system triggers a behavior of vital importance: escaping from predators. A combination of fast afferent circuits, synaptic specialization, special intrinsic properties, and topology of the M-cell contribute in concert to a fast startle escape response (*Eaton et al., 1991*; *Korn and Faber, 2005*). Here, we show that the M-cell incorporates multisensory information to enhance its responses and characterize how the integration of multiple stimuli is affected by the modality, spatial source, temporal structure, temporal order, and intensity of the stimuli. Although multisensory enhancement has been previously described in different neural populations and across a wide range of species, it was unclear whether the activity of these neurons was necessary, sufficient, or even influenced the execution of behavioral responses to multisensory stimuli. In contrast to other known multisensory structures of vertebrates MSI in the M-cell has an unequivocal function in fish behavior and survival, as it directly determines whether and when a C-start is triggered (*McIntyre and Preuss, 2019*; *Martorell and Medan, 2022*).

Different factors differentially affect the summation and integration of auditory and visual inputs to the M-cell. First, sensory inputs are subject to different passive decays depending on the site of the dendritic arbor at which they arrive. Auditory inputs reach the M-cell distal lateral dendrite through dense mixed electrical and chemical synapses driven through the posterior eighth nerve afferents and the proximal dendrite through indirect glutamatergic inputs conveying information from inner ear endorgans (*Szabo et al., 2006*; *Szabo et al., 2007*). In contrast, visual inputs arriving to the ventral dendrite establish synapses that do not show additional specializations as the club endings (*Zottoli et al., 1987*; *Cachope et al., 2007*; *Szabo et al., 2007*). Second, the geometrical properties of the M-cell lateral and ventral dendrites (length and tapering) impose different filters on the input signal which determine a stronger spatial decay for visual than for auditory signals (*Medan et al., 2018*).

Third, as excitatory (and inhibitory) signals reach the soma, the active ion conductances decrease the membrane resistance and impact the response to following signals (*Hodgkin, 1947*). Finally, the magnitude and dynamics of the stimuli will also affect integration. Both abrupt auditory pips and longer lasting auditory looms generate fast EPSPs that depend on the stimulus frequency and a depolarizing envelope proportional to the stimulus amplitude (*Szabo et al., 2006*) that slowly decays to baseline (*Figure 4*). Although the pathways for auditory and visual inputs are completely different, tectal responses also consist of fast peaks that track stimulus frequency and a tonic depolarizing envelope proportional both to stimulus duration and frequency (*Figure 1*) that also decays slowly (*Figure 4*).

These temporal dynamics of the responses modulate the effective MSI in the M-cell soma, which emerges as a complex, nonlinear interaction between 'positive' (input strength and summation, voltage-dependent depolarizing conductances) and 'negative' (temporal and spatial decay, FFI, and depolarization-induced decrease in input resistance) contributions. These factors likely influence integration in any multisensory neuron. However, the accumulated knowledge on the presynaptic pathways and the intrinsic mechanisms operating in the M-cell combined with its functional role in commanding the initiation of the C-start response makes this neuron a unique model for analyzing MSI in a single identifiable neuron.

We have previously shown that fish exhibit a behavioral multisensory enhancement of the C-start escape response for audiovisual stimuli, and that this effect is reduced when increasing the salience of the unisensory stimuli (*Martorell and Medan, 2022*). Therefore, one of our goals here was to investigate if the inverse effectiveness principle observed behaviorally had its correlation in integration processes taking place in the M-cell. Paralleling our previous behavioral results, here we show through intracellular recordings that the response evoked by an auditory pip can be increased by about 50% when it is preceded by a tectal stimulus (*Figure 2C, G*). This enhancement was not affected by relatively large changes in stimulus frequency. The plateau in the tonic depolarization observed 30–50 ms after a sustained stimulus (*Figure 1A*) combined with the low-pass filtering properties of the membrane could be responsible for the lack of frequency effects. These two mechanisms point to the M-cell role as a 'sudden' stimulus detector rather than a precise coder of sensory stimulus properties.

When we combined a tectal train with an auditory pip to obtain a multisensory stimulus, the response was smaller than the sum of the evoked depolarization of the two components (*Figure 2D, H*). This sublinear effect was expected in the case of passive integration, since cellular biophysics predict a sublinear integration of subthreshold inputs due to a reduction in the driving force of sodium and the aperture of ion channels that reduce the membrane resistance. Noteworthy, action potentials are all-or-nothing events, so the effects when the membrane potential is close to the threshold, sublinear summation can determine that an otherwise silent neuron fires an AP (*Figure 5C*). Therefore, a weak (subthreshold) auditory pip combined by a weak visual stimulus in close temporal association might be sufficient for triggering an escape response. Indeed, analysis of 60 M-cells tested with A, tectal and multimodal stimuli showed that auditory stimuli evoked firing in 5% of cells, tectal stimuli in 6.7% of cells but almost 12% of M-cells fired in response to multisensory stimuli. Even though these results are probably an underestimation of actual firing probabilities due to anesthesia and paralyzing agents, they provide a mechanistic support to our previous behavioral observations (*Martorell and Medan, 2022*).

Importantly, for the weakest multisensory stimuli the M-cell response deviated from this sublinear integration (*Figures 2C and 5A*). When the sum of unisensory components is less than 2.5 mV (essentially a single tectal stimulus immediately followed by a single auditory pip) summation is linear (*Figures 2D and 5A*). This suggests the existence of one or more active mechanism/s of MSI upstream of the M-cell soma, potentially through a presynaptic neuron that also receives audiovisual inputs and projects to the M-cell or locally at the level of the dendritic tree of the M-cell. Moreover, previous work has described the M-cell membrane nonlinearities in the lateral and ventral dendrites which progressively increase M-cell excitability for depolarizations above 5 mV (*Faber and Korn, 1986*; *Medan et al., 2018*). Such nonlinear properties might enhance MSI of weak stimuli (*Martorell and Medan, 2022*) providing a cellular mechanism for the principle of inverse effectiveness of MSI.

The temporal structure of the multisensory stimuli is crucial to allow the animal to effectively 'bind' or associate both signals as coming from a single source (*Stein and Stanford, 2008*). In the M-cell, this translates into the time window where the effect on the membrane voltage (excitatory or inhibitory)

persists. Here, we found that a delay of 50 ms between the components of a low-amplitude multisensory stimuli result in a linear summation of the signals (*Figure 4C*) and that the temporal order of the components produces a slight effect in the integration (*Figure 4C*, AT vs. TA) (*Felch et al., 2016*). As it has been previously suggested, the lengthy decay of the slow EPSP of the auditory pip (or the tonic component of the tectal train) probably contributed to relax the requirement for coincident inputs, effectively broadening the temporal window of integration (*Szabo et al., 2006*). On the other hand, the integration of unisensory stimuli was dependent on the modality of the stimuli: two tectal (TT) stimuli summed linearly while two auditory (AA) stimuli summed sublinearly (*Figure 4C*). This difference might be due to differences in both the presynaptic networks that mediate PPI (*Tabor et al., 2018*) and in the temporal decay of auditory and tectally evoked FFI (*Figure 3D*). Auditory stimuli produce a longer-lasting inhibition than tectal stimuli of similar strength (*Figure 3F*) and while tectal inhibition is almost extinct 50 ms after a tectal stimulus, auditory inhibition stabilizes at around 50% of the maximum inhibitory effect in a time window of 30–70 ms after an auditory stimulus (*Figure 3E*). Interestingly, the integration of AT stimuli was smaller than that of TA, supporting a differential cross-modal and intra-modal integration of the excitatory or inhibitory components (*Figure 4C*). Another factor that could be contributing to the differential integration between AA and AT stimuli is that FFI inhibition is likely to be heterogeneously spread in the perisomatic area around the M-cell. If auditory stimulation recruits inhibitory PHP neurons that target the proximal lateral dendrite of the M-cell, a subsequent sensory input propagating along the lateral dendrite (i.e, auditory input) would suffer stronger attenuation than one reaching the soma through the ventral dendrite (i.e. tectal input). However, specific studies are required to test this hypothesis as it is still unclear if auditory and tectal afferent fibers synapse the same or different sets of inhibitory neurons and which is the spatial distribution of those inhibitory contacts. Additionally, presynaptic elements could also contribute to the asymmetry in unimodal auditory (AA) vs. unimodal tectal (TT) or multisensory processing. A subset of glutamatergic Gsx1 neurons located in rhombomere 4 have been shown to mediate M-cell auditory presynaptic inhibition (*Tabor et al., 2018*). These neurons produce presynaptic inhibition of eighth nerve auditory fibers adjacent to the distal portion of the M-cell lateral dendrite but leave unaffected the ventral dendrite. This mechanism could therefore affect the processing of pairs of sensory signals that initiate with an auditory component (AA, AT) and leave the integration of tectal or multisensory stimuli unaffected.

## Materials and methods

### Animals

We used adult goldfish (*Carassius auratus*) of both sexes and 7–10 cm of standard body length. Fish were purchased from Billy Bland Fishery (Taylor, AR, USA), Hunting Creek Fisheries (Thurmont, MD, USA) or Daniel Corriarello Aquarium (Buenos Aires, Argentina) and allowed to acclimate for at least a week after transport. Fish were kept in rectangular Plexiglas holding tanks (30 × 60 × 30 cm$^3$; 54 l) in groups of up to 10 individuals. Tanks were supplied with filtered and dechlorinated water and maintained at 18°C. Ambient light was set to a 12-hr light/dark photoperiod. Animals were fed floating pellets (Sera, Germany) five times a week. All procedures and protocols were performed in accordance with the guidelines and regulations of the Institutional Animal Care and Use Committee of Hunter College, City University of New York and Facultad de Ciencias Exactas y Naturales, Universidad de Buenos Aires.

### Auditory stimuli

Sound stimuli consisted in single-cycle sound pips (5 ms duration at 200 Hz) produced by a function generator (33210A Agilent Technologies Inc, Santa Clara, CA, USA) connected to a shielded subwoofer (SA-WN250 Sony Corp., Tokyo, Japan) located 30 cm to the right from the recording chamber. Sound stimuli were recorded with a microphone (XM8500, Behringer) positioned 10 cm over the head of the fish. Calibration recordings were performed with a hydrophone (SQ01, Sensor Technology, Ontario, Canada) positioned where the submerged fish was placed during experiments. Since we were interested in studying integration of subthreshold stimuli, we used sound intensities known to be subthreshold for evoking behavioral startle responses (*Neumeister et al., 2008*; *Weiss et al., 2009*; *Martorell and Medan, 2022*). For all experiments we used a sound intensity of 118 dB SPL

(relative to 1 µPa in water), except for the experiments in *Figure 4* where we used a sound intensity of 147 dB SPL (relative to 1 µPa in water). Post hoc analysis of the M-cell responses in this study revealed that only 3 out of 60 M-cells (5%) fired at least once to an auditory stimulus.

## Optic tectum stimulation

Tectal stimuli were administered to the mid-anterior rim of the left tectum using a bipolar electrode (#30202 FHC, Bowdoin, ME, USA) connected to an isolated stimulator (DS2A; Digitimer Ltd, Welwyn Garden City, UK). In most cases, stimulation [100 µs, 0.1–1 mA] in this area was effective in driving subthreshold responses (<4 mV) in the M-cell. To change the strength and dynamics of the tectal stimulation we either used a 60-Hz stimulation train of variable duration (1, 33, 66, 100, and 200 ms) or a 100-ms train of variable frequency (1, 33, 100, and 200 Hz). We deliberately used tectal stimuli that evoked M-cell firing with a very low probability (6.7%, 4 out of 60 M-cells tested fired at least once to any of the tectal stimuli used).

## Electrophysiology

M-cell intracellular responses to tectal and acoustic stimuli were studied in vivo using standard surgical and electrophysiological recording techniques (*Preuss and Faber, 2003*; *Preuss et al., 2006*; *Medan et al., 2018*). To initiate anesthesia, fish were immersed in 1 l of ice water with 40 mg/l of the general anesthetic tricaine methanesulfonate (MS-222, Western Chemical, Ferndale, WA, USA), until the fish ceased to swim, lost equilibrium and were unresponsive to a pinch on the tail (typically 10–15 min). They were next treated with 20% benzocaine gel (Ultradent, South Jordan, UT, USA) at incision sites and pin-holding points 5 min prior to surgical procedures. Fish were stabilized in the recording chamber by two pins, one on each side of the head, and ventilated through the mouth with recirculating, aerated saline at 18°C (saline [g/l]: sodium chloride 7.25, potassium chloride 0.38, monosodium phosphate monobasic 0.39, magnesium sulfate 0.11, HEPES (N-2-hydroxyethylpiperazine-N'-2-ethanesulfonic acid) 4.77; calcium chloride 0.24; dextrose 1.01, pH 7.2). The recording chamber was mounted inside an opaque, thin-walled tank filled with saline that covered the fish up to eye level. The recirculating saline also included a maintenance concentration of the anesthetic MS-222 (20 g/l) that does not interfere with auditory processing (*Palmer and Mensinger, 2004*; *Cordova and Braun, 2007*). Next, the spinal cord was exposed with a small lateral incision at the caudal midbody. Bipolar stimulation electrodes were placed on the unopened spinal cord to transmit low-intensity (5–8 V) electrical pulses generated by an isolated stimulator (A 360, WPI, Sarasota, FL, USA). This allowed antidromic activation of the M-cell axons, as confirmed by a visible muscular contraction (twitch). Surgical procedures were performed before a muscle paralysis agent was injected, which allows monitoring the effectiveness of the anesthetic by watching for an increase of opercula movement frequency (largely reduced in deep anesthesia) and movements/twitches in response to the surgical procedures. Shortly before the recordings started, animals were injected I.M. with D-tubocurarine (1 µg g$^{-1}$ b.w.; Abbott Laboratories, Abbott Park, IL, USA) and a small craniotomy exposed the medulla for electrophysiological recordings. Antidromic stimulation produces a negative potential in the M-cell axon cap (typically 15–20 mV), which unambiguously identifies the axon hillock and allows intracellular recordings from defined locations along the M-cell soma-dendritic membrane (*Furshpan and Furukawa, 1962*; *Furukawa, 1966*; *Faber and Korn, 1989*). Intracellular recordings were acquired using borosilicate glass electrodes (7–10 MΩ) filled with 5 M potassium acetate and an Axoclamp-2B amplifier (Axon Instruments, Fostefvr City, CA, USA) in current-clamp setting. M-cell responses were acquired with a Digidata 1440A (Axon Instruments) at 25 kHz. Electrodes were advanced using motorized micromanipulators (MP-285; Sutter Instruments, Novato, CA, USA) until reaching the axon cap (defined as a site with an extracellular M-cell AP field >10 mV). Next, the electrode was moved 50 µm lateral and 50 µm posterior to penetrate the somatic region. Only trials in which the resting membrane potential was between −90 and −70 mV were included in the analysis.

## Statistical analyses and experimental design

To perform statistical analysis, we used R (version 4.0.2, https://www.r-project.org) and RStudio (version 1.1.456, www.rstudio.com). A significance level of $\alpha$ = 0.05 was used throughout the study. Effects of explanatory variables over response variables were assessed using linear models followed by ANOVA or binomial generalized linear models in the case of binary variables. Paired *T*-tests were

used to compare tectal or auditory responses recorded in left or right M-cells (see *Figure 1*). Two tailed Wilcoxon ranked sum tests were used for single sample comparisons. Sample size is denoted by *N* when it refers to the number of goldfish, *n* when it refers to the number of trials (average of 5–10 presentations). In experiments where we recorded from both M-cells of the same animals, results were pooled. Boxplots show the median, 25th and 75th percentiles, and lines indicate minimum and maximum values.

Postsynaptic responses to acoustic and tectal stimuli were recorded at the M-cell soma and averages of 5–10 responses were used in the statistical analysis. M-cell responses to trains of tectal stimulation consist of phasic postsynaptic potentials riding on top of a tonic depolarizing response (*Figure 1A–C*). We analyzed the contribution of the phasic and tonic components separately (*Figure 1A'*). To do this we considered a 12-ms window following the last tectal pulse (color-shaded areas in *Figure 1A–C*) and defined the tonic response amplitude as the difference between the membrane potential immediately before the arrival of the last pulse (blue bar in *Figure 1A'*) and resting membrane potential. The phasic response was defined as the difference between the peak voltage in response to the last train pulse and the depolarization reached immediately before the last tectal pulse (orange bar in *Figure 1A'*).

To characterize MSI, we quantified responses calculating the mean evoked depolarization in a 12-ms window after sound onset (auditory only trials, A) or after the last tectal stimulus in tectal-only (tectal only trials, T) and multisensory (tectal + auditory trials, M) trials (*Figure 2*), considering both phasic and tonic components. We also measured the peak amplitude of the response, which in all cases occurred in the same time window. As peak amplitude analysis resulted in the same qualitative results, only the mean area analysis is shown.

For quantification, the responses to multimodal stimuli were normalized to the tectal and auditory components. In the field of MSI, multisensory responses are typically normalized to the maximum response of the unisensory stimulus (*Meredith and Stein, 1983*; *Stanford and Stein, 2007*; *Stein and Stanford, 2008*), which we call Multisensory-Maximum Unimodal index (MSI/Max):

$$MSI/Max = M \div max\,(T,\,A)$$

The value of this index determines the net effect of the integration, where *M* is the multisensory response and *max(T, A)* is the largest of the two unisensory components (all evaluated as the area in a 12-ms window as defined above). Therefore, an *MSI/Max* greater than 1 represents multisensory enhancement, *MSI/Max* lower than 1 represents multisensory inhibition, and *MSI/Max* not significantly different than 1 suggests absence of MSI (*Stein and Stanford, 2008*). Additionally, to calculate the linearity of the MSI performed by the M-cell, we calculated the ratio between the multisensory response and the sum of the unisensory responses, which we call Multisensory-Sum Unimodals index (MSI/Sum):

$$MSI/Sum = M \div (T + A)$$

The value of this index indicates the linearity of the integration, determined by the ratio of the multisensory response *M* and the sum of the unisensory components, tectal (*T*) and auditory (*A*). *MSI/Sum* greater than 1 representing supralinear integration, *MSI/Sum* of 1 representing linear integration, and *MSI/Sum* lower than 1 representing sublinear integration (*Stanford et al., 2005*).

The M-cell is subject to FFI which can be triggered by either auditory or tectal stimuli (*Faber and Korn, 1982*; *Medan et al., 2018*). Evoked FFI can be quantified as the amplitude reduction of an antidromically evoked 'test' action potential (APtest) after presenting a single auditory pip or a brief tectal train as 'conditioning' stimuli of similar strength. Both stimuli evoked comparable peak depolarizations in the M-cell (~5 mV). To test if FFI evoked by auditory or tectal stimuli differed, we evoked the APtest at specific times (2–80 ms) following a sensory stimulus and calculated shunting inhibition as %SI = 100 − APtest/APcontrol × 100, where AP control is the amplitude of the AP without a conditioning stimulus (*Figure 3A*).

Our dataset (raw traces of electrophysiological recordings of the Mauthner cell) and scripts are deposited in Dryad (DOI: 10.5061/dryad.rxwdbrvkj).

## Acknowledgements

This work was supported by PICT 2012-1578 and PICT 2017-0007, FONCYT-ANPCYT (VM), PIP 11220130100729CO (VM), UBACyT 20020130300008BA (VM), Thalmann Program, University of Buenos Aires (VM) SOC was supported by undergraduate fellowships from the National Interuniversity Council and University of Buenos Aires. The authors thank Dr. Lidia Szczupak and members of the Medan lab for discussion and for reading previous versions of this manuscript. The current address of SOC is at The Rockefeller University, New York, NY, USA (10065).

## Additional information

### Funding

| Funder | Grant reference number | Author |
| --- | --- | --- |
| Fondo para la Investigación Científica y Tecnológica | PICT 2012-1578 | Violeta Medan |
| Fondo para la Investigación Científica y Tecnológica | PICT 2017-0007 | Violeta Medan |
| Fondo para la Investigación Científica y Tecnológica | FONCYT-ANPCYT | Violeta Medan |
| University of Buenos Aires | Thalmann Program | Violeta Medan |
| Consejo Nacional de Investigaciones Científicas y Técnicas | 11220130100729CO | Violeta Medan |
| University of Buenos Aires | Undergraduate Fellowship | Santiago Otero-Coronel |

The funders had no role in study design, data collection, and interpretation, or the decision to submit the work for publication.

### Author contributions

Santiago Otero-Coronel, Conceptualization, Data curation, Formal analysis, Investigation, Methodology, Writing – original draft, Writing – review and editing, Software, Visualization; Thomas Preuss, Conceptualization, Resources, Methodology, Writing – original draft, Writing – review and editing; Violeta Medan, Conceptualization, Resources, Data curation, Formal analysis, Funding acquisition, Investigation, Methodology, Writing – original draft, Project administration, Writing – review and editing, Supervision, Visualization

### Author ORCIDs

Santiago Otero-Coronel https://orcid.org/0000-0003-4633-7111
Violeta Medan https://orcid.org/0000-0002-3612-0659

### Ethics

All procedures and protocols were performed in accordance with the guidelines and regulations of the Institutional Animal Care and Use Committee of Hunter College, City University of New York and Facultad de Ciencias Exactas y Naturales protocols (#52), Universidad de Buenos Aires.

Reviewer #1 (Public Review): https://doi.org/10.7554/eLife.99424.4.sa1
Reviewer #2 (Public Review): https://doi.org/10.7554/eLife.99424.4.sa2
Author response https://doi.org/10.7554/eLife.99424.4.sa3

## Additional files

### Supplementary files
• MDAR checklist

### Data availability
All data generated or analyzed during this study are included in the manuscript. Our dataset (raw traces of electrophysiological recordings of the Mauthner cell) and MATLAB scripts to analyze data are available at Dryad, https://doi.org/10.5061/dryad.rxwdbrvkj.

The following dataset was generated:

| Author(s) | Year | Dataset title | Dataset URL | Database and Identifier |
| --- | --- | --- | --- | --- |
| Medan V, Otero Coronel S, Preuss T | 2024 | Multisensory integration enhances audiovisual responses in the Mauthner cell | http://dx.doi.org/10.5061/dryad.rxwdbrvkj | Dryad Digital Repository, 10.5061/dryad.rxwdbrvkj |

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
