## [Editor Report · eLife Assessment]

This study provides **valuable** advances in our understanding of how inputs from multiple sources can impact the physiology of motor neurons during the process of multisensory integration. Specifically, the authors show how streams of auditory and principally visual information modulate the physiology of Mauthner neurons in goldfish, thus allowing the different senses to influence escape behavior. Supporting evidence is generally **convincing**, although material reporting the direct control of behavior is less representative of the data.

---

## [Referee Report · Reviewer #1 (Public Review)]

Otero-Coronel et al. address an important question for neuroscience - how does a premotor neuron capable of directly controlling behavior integrate multiple sources of sensory inputs to inform action selection? For this, they focused on the teleost Mauthner cell, long known to be at the core of a fast escape circuit. What is particularly interesting in this work is the naturalistic approach they took. Classically, the M-cell was characterized, both behaviorally and physiologically, using an unimodal sensory space. Here the authors make the effort (substantial!) to study the physiology of the M-cell taking into account both the visual and auditory inputs. They performed well-informed electrophysiological approaches to decipher how the M-cell integrates the information of two sensory modalities depending on the strength and temporal relation between them.

The empirical results are convincing and well-supported. The manuscript is well-written and organized. The experimental approaches and the selection of stimulus parameters are clear and informed by the bibliography. The major finding is that multisensory integration increases the certainty of environmental information in an inherently noisy environment.

---

## [Referee Report · Reviewer #2 (Public Review)]

In this manuscript, Otero-Coronel and colleagues use a combination of acoustic stimuli and electrical stimulation of the tectum to study MSI in the M-cells of adult goldfish. They first perform a necessary piece of groundwork in calibrating tectal stimulation for maximal M-cell MSI, and then characterize this MSI with slightly varying tectal and acoustic inputs. Next, they quantify the magnitude and timing of FFI that each type of input has on the M-cell, finding that both the tectum and the auditory system drive FFI, but that FFI decays more slowly for auditory signals. These are novel results that would be of interest to a broader sensory neuroscience community. By then providing pairs of stimuli separated by 50ms, they assess the ability of the first stimulus to suppress responses to the second, finding that acoustic stimuli strongly suppress subsequent acoustic responses in the M-cell, that they weakly suppress subsequent tectal stimulation, and that tectal stimulation does not appreciably inhibit subsequent stimuli of either type. Finally, they show that M-cell physiology mirrors previously reported behavioural data in which stronger stimuli underwent less integration.

The manuscript is generally well-written and clear. The discussion of results is appropriately broad and open-ended. It's a good document. Our major concerns regarding the study's validity are captured in the individual comments below. In terms of impact, the most compelling new observation is the quantification of the FFI from the two sources and the logical extension of these FFI dynamics to M-cell physiology during MSI. It is also nice, but unsurprising, to see that the relationship between stimulus strength that MSI is similar for M-cell physiology to what has previously been shown for behavior. While we find the results interesting, we think that they will be of greatest interest to those specifically interested in M-cell physiology and function.

---

## [Author Response]

The following is the authors’ response to the previous reviews.

Minor Concern (Original Comment 1):“We think that this is sufficient to address our concern. Some citations may be in order to underpin the new text.”

We appreciate the reviewer’s assessment that the revised text clarifies the complexity of the upstream circuitry beyond the retina, including inputs from the thalamus. As recommended, we have now included additional citations in the revised manuscript to support these points.

Major Concern (Original Comment 5):“We do not feel that this important concern has been addressed. The stats are definitively negative. There is no statistical evidence from these data that multisensory integration is occurring in this assay. The anesthesia, paralysis, and low n may provide explanations for this negative result, but it is still a negative result (p=0.5269). To show two examples of multisensory integration for subthreshold stimuli fits the narrative, but this result is not supported. Examples where individual stimuli caused APs (and combined stimuli did not) also occurred, presumably, and at a rate that is statistically indistinguishable to the examples shown in Figure 5. As such, if results from this assay are going to be in the manuscript, acoustic-only and tectum-only examples should be shown as well, although they would not fit the narrative. To be meaningful, this experiment would have to show that multisensory integration is happening in this circuit. Frustrating though it must be, the experiment has given a negative result to that question.”

We understand the reviewer’s concern regarding Figure 5C and the firing of action potentials (APs) in response to multisensory stimuli. We acknowledge that our assay is not suited to answer this question definitively and that our results do not provide statistical support for this hypothesis. In response, we have removed the examples previously shown in Figure 5C, along with the related description in the Results section (lines 420–426), to avoid implying unsupported integration in suprathreshold conditions.